# Glucose inhibits cardiac muscle maturation through nucleotide biosynthesis

Haruko Nakano[1], Itsunari Minami[2], Daniel Braas[3], Herman Pappoe[1], Xiuju Wu[4], Addelynn Sagadevan[1], Laurent Vergnes[5], Kai Fu[1], Marco Morselli[1], Christopher Dunham[6], Xueqin Ding[6], Adam Z Stieg[7,8], James K Gimzewski[6,7,8,9], Matteo Pellegrini[1,9,10], Peter M Clark[3,9,11], Karen Reue[5,10], Aldons J Lusis[4,5,10,12], Bernard Ribalet[13], Siavash K Kurdistani[9,10,14,15], Heather Christofk[3,10,14,15], Norio Nakatsuji[2,16], Atsushi Nakano[1,9,10,15]*

[1]Department of Molecular, Cell, and Developmental Biology, University of California, Los Angeles, Los Angeles, United States; [2]Institute for Integrated Cell-Material Sciences (WPI-iCeMS), Kyoto University, Kyoto, Japan; [3]Department of Molecular and Medical Pharmacology, University of California, Los Angeles, Los Angeles, United States; [4]Division of Cardiology, Department of Medicine, University of California, Los Angeles, Los Angeles, United States; [5]Department of Human Genetics, University of California, Los Angeles, Los Angeles, United States; [6]Department of Chemistry and Biochemistry, University of California, Los Angeles, Los Angeles, United States; [7]California NanoSystems Institute, University of California, Los Angeles, Los Angeles, United States; [8]WPI Center for Materials Nanoarchitectonics (MANA), National Institute for Materials Science, Meguro, Japan; [9]Jonsson Comprehensive Cancer Center, University of California, Los Angeles, Los Angeles, United States; [10]Molecular Biology Institute, University of California, Los Angeles, Los Angeles, United States; [11]Crump Institute for Molecular Imaging, University of California, Los Angeles, Los Angeles, United States; [12]Department of Microbiology, Immunology, and Molecular Genetics, University of California, Los Angeles, Los Angeles, United States; [13]Department of Physiology, University of California, Los Angeles, Los Angeles, United States; [14]Department of Biological Chemistry, University of California, Los Angeles, Los Angeles, United States; [15]Eli and Edythe Broad Center of Regenerative Medicine and Stem Cell Research, University of California, Los Angeles, Los Angeles, United States; [16]Institute for Life and Frontier Medical Sciences, Kyoto University, Kyoto, Japan

*For correspondence: anakano@ucla.edu

**Abstract** The heart switches its energy substrate from glucose to fatty acids at birth, and maternal hyperglycemia is associated with congenital heart disease. However, little is known about how blood glucose impacts heart formation. Using a chemically defined human pluripotent stem-cell-derived cardiomyocyte differentiation system, we found that high glucose inhibits the maturation of cardiomyocytes at genetic, structural, metabolic, electrophysiological, and biomechanical levels by promoting nucleotide biosynthesis through the pentose phosphate pathway. Blood glucose level in embryos is stable in utero during normal pregnancy, but glucose uptake by fetal cardiac tissue is drastically reduced in late gestational stages. In a murine model of diabetic pregnancy, fetal hearts showed cardiomyopathy with increased mitotic activity and decreased maturity. These data suggest that high glucose suppresses cardiac maturation, providing a possible mechanistic basis for congenital heart disease in diabetic pregnancy.

DOI: https://doi.org/10.7554/eLife.29330.001

## Introduction

Congenital heart disease (CHD) is the most common type of birth defect affecting 0.8% of human live births (*Fahed et al., 2013*). Although genetic factors play a significant role in the development of CHD, current genomic technologies, including exome sequencing and SNP arrays, have provided a genetic diagnosis for only 11% of the probands (*Gelb et al., 2013*), highlighting the crucial role of non-genetic contributors.

Among the non-genetic factors that influence the fetal heart, maternal hyperglycemia is the most common medical condition, associated with a 2–5-fold increase in CHD independent of genetic contributors (*Centers for Disease Control, 1990*; *Simeone et al., 2015*; *Yogev and Visser, 2009*). Diabetic pregnancy is often accompanied by maternal complications including vasculopathy, neuropathy, and insulin resistance, which potentially affect fetal cardiac formation indirectly. These systemic complications are often subclinical, hindering the dissection of the pathomechanism of CHD in diabetic pregnancy. Thus, despite the established association between maternal hyperglycemia and malformation of the fetal heart, little is known about how glucose levels impact cardiomyocyte development and how hyperglycemia affects heart formation in diabetic pregnancy (*Gaspar et al., 2014*).

The metabolic environment is one potential non-genetic determinant of cell proliferation and differentiation. Cells display distinct metabolic characteristics depending on their differentiation stage (*Carey et al., 2015*; *Tohyama et al., 2016*; *Wang et al., 2009*), and the fuel type used by the cells serves not merely as a source of energy but also as a critical regulator of self-renewal and differentiation of stem or progenitor cells (*Harris et al., 2013*; *Oburoglu et al., 2014*; *Shiraki et al., 2014*; *Shyh-Chang et al., 2013*). However, little is known about the mechanism. Cardiomyocytes shift their energy substrate during late embryonic and neonatal stages (*Makinde et al., 1998*). Glucose is the major energy source during the early developmental stages. Oxidative phosphorylation is low until E10.5 of developing rodent hearts and rapidly increases between E10.5 and E14.5 (*Cox and Gunberg, 1972*). This coincides with the rapid maturation of the mitochondrial structure in embryonic cardiomyocytes (*Mackler et al., 1971*). Shortly after birth, fatty acid oxidation becomes the predominant source of ATP production to meet the high energy demand of the maturing heart (*Warshaw and Terry, 1970*). These metabolic changes occur as a consequence of changes in the expression of metabolic enzymes and transporters. However, it remains unclear whether and how these metabolic changes, in turn, regulate the cardiac differentiation program.

Here, we describe the use of in vitro human embryonic stem cell-derived cardiomyocytes (hESC-CMs) and an in vivo murine diabetic model to show that glucose not only induces cardiomyocyte proliferation but also inhibits cardiomyocyte maturation. Glucose can be metabolized in multiple catabolic and anabolic pathways, including glycolysis, oxidative phosphorylation, the pentose phosphate pathway (PPP) and the hexosamine biosynthesis pathway. Our chemical screening revealed that the pro-mitotic and anti-maturation effect of high glucose is regulated by glucose-derived deoxynucleotide biosynthesis through PPP. In vivo measurement of [18]F-FDG accumulation revealed that glucose uptake is drastically suppressed during the late gestational and early postnatal stages. Exposure to high blood glucose in a murine model of diabetic pregnancy resulted in higher mitosis and delayed maturation of fetal cardiomyocytes in vivo. Together, our data uncover how the dynamics of glucose metabolism impact late embryonic cardiogenesis.

## Results

### Glucose reduction promotes hESC-CM differentiation

hESC-CMs were differentiated in monolayer in a chemically defined condition, reproducibly yielding ~90% of MF20[+] cardiomyocytes at day 14 with multiple cell lines including WA09 (H9) and UCLA4 hESCs (*Figure 1A–C*) (*Arshi et al., 2013*; *Minami et al., 2012*). hESC-CMs start to beat synchronously at around day 6–7 in our system. To characterize the differentiation stages, mRNA expression profiles from H9 hESC-CMs were serially examined by RNA-seq at five distinct stages (GSE84815): undifferentiated hESC (day 0); mesodermal precursor stage (hMP, day 2); cardiac

**eLife digest** Congenital heart disease is the most common type of birth defect, affecting nearly 1 in 100 children born. It can involve a weak heart, narrowed arteries, narrowed heart valves, or the main arteries of the heart switching places. These conditions can be fatal if untreated and often need surgery to correct.

The mother's blood sugar levels during pregnancy can have a large effect on how likely the baby is to have congenital heart disease. If a pregnant woman has poorly controlled diabetes with rapidly fluctuating sugar levels, she may be at a higher risk of having a child with the condition. High sugar levels in the mother's blood make the baby up to five times more likely to have congenital heart disease. It has been difficult to find out exactly how sugar levels interfere with heart development because diabetes can affect the fetus in many ways.

Nakano et al. used stem cells and experiments in pregnant mice with diabetes to hone in on how high sugar levels affect the fetus's heart development. First, heart cells were grown from human stem cells, and exposed to high levels of glucose in a dish. This revealed a new mechanism for how high sugar levels affect heart formation: the cells created too many nucleotides, the building blocks of molecules such as DNA. It turns out that high glucose levels boosted a chemical process in the cell known as the pentose phosphate pathway. Some of the products of this pathway are nucleotides. This made the cells divide rapidly, but did not allow them to mature well compared with cells exposed to normal levels of sugar. In another experiment, Nakano et al. found similar results in pregnant diabetic mice. The heart cells in mouse fetuses also divided quickly but matured slowly when exposed to high sugar levels.

An estimated 60 million women at an age to have children have diabetes. These new findings help us to understand why and how these women are more likely to have children with congenital heart disease, and further study will hopefully lead to a better way to prevent this condition.

DOI: https://doi.org/10.7554/eLife.29330.002

progenitor stage (hCP, day 5); immature cardiomyocyte (hCM14); and hESC-CMS differentiated for 14 additional days (hCM28). The expression data were analyzed using signatures collected from MSigDB, a body atlas and primary cell atlas (*Mabbott et al., 2013*; *Su et al., 2004*; *Subramanian et al., 2005*). As expected, the stem cell signature decreases during these five stages, while signatures associated with heart and smooth muscle increase, further suggesting that our protocol leads to a high degree of enrichment for cardiomyocytes (*Figure 1D and E*). This differentiation course is comparable to that reported in previous publications (*Paige et al., 2012*; *Wamstad et al., 2012*).

To examine the impact of glucose levels on cardiac differentiation, hESC-CMs were cultured in media containing various concentration of glucose, starting at the hCM14 stage when cells are already differentiated to immature cardiomyocytes (*Figure 2A*, *Video 1* and *2*). The basal differentiation medium contains 25 mM glucose, 0.9 mM pyruvate, essential and nonessential amino acids, and human albumin (G25 medium). Interestingly, glucose dose-dependently suppressed the expression of *TNNT2* (a cardiac marker), *NKX2-5* (a cardiac marker), and *PPARGC1A* (a mitochondrial marker) (*Figure 2B*). Gene expression profiling by RNA-seq revealed that genes that are related to cardiac muscle and function are enriched in hESC-CMs in low glucose medium, and that genes that are associated with mitosis and cell cycle are enriched in the high-glucose group genome-wide (*Figure 2C*, *Figure 2—figure supplement 1B*; GSE84814). These data suggest that low glucose after day 14 induces the differentiation and suppresses the cell cycle of hESC-CMs.

To validate these results, hESC-CM proliferation was analyzed by pH3 staining and EdU flow cytometry analysis. Low glucose decreased mitotic activity at day 28 without affecting the viability of hESC-CMs (*Figure 2D,E*). In addition, hESC-CMs in low glucose medium showed more robust staining of α-actinin, although the sarcomere length did not significantly change (*Figure 2F*, *Figure 2—figure supplement 1B*). MitoTracker staining and flow cytometry analyses revealed that hESC-CMs cultured in low glucose media have increased mitochondrial contents and inter-myofibrillar distribution of mitochondria, characteristic of differentiated cardiomyocytes (*Figure 2F and G*). Addition of 2-DG (2-deoxy-D-glucose), a competitive inhibitor of glucose phosphorylation, induced higher levels

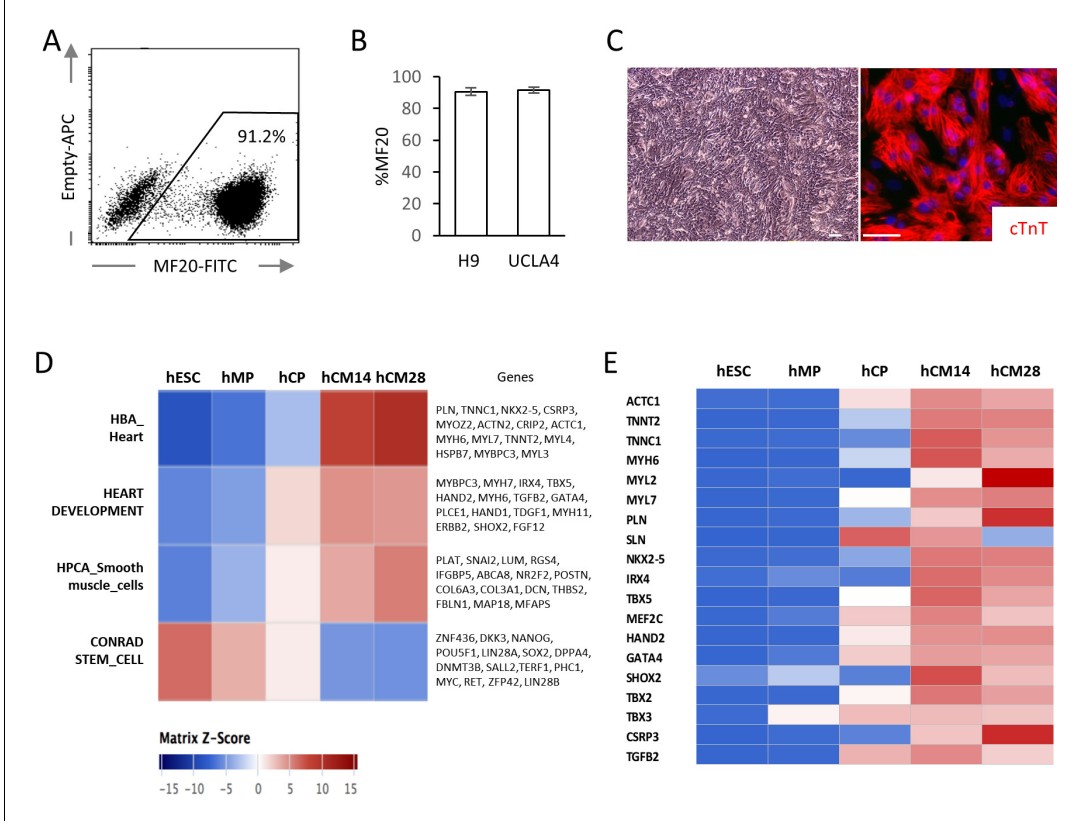

**Figure 1.** High yield hESC-CMs derived under chemically defined conditions recapitulate developmental time course. (**A**) Representative flow cytometry analysis of a cardiac marker, MF20 (myosin) in hESC-CMs (H9) at day 14. (**B**) Bar chart of the percentage of MF20$^+$ cardiomyocytes in hESC-CMs generated from H9 (n = 8) and UCLA4 cells (n = 3, mean ±S D), as determined by flow cytometry analysis. (**C**) Images of phase-contrast and Tnnt2 (cTnT: cardiac Troponin T) immunofluorescent staining of hESC-CMs at day 28. Scale bar = 50 μm. (**D**) The time course of the gene expression profile obtained by RNA-seq using MSigDB shown as a heatmap (n = 3). hCM14 or hCM28, human ESC-derived cardiomyocyte differentiation day 14 or 28; hCP, human cardiac progenitor; hESC, human embryonic stem cell; hMP, human mesodermal precursor. (**E**) Heatmap of representative cardiac genes from (**D**), showing progressive upregulation over the time.

DOI: https://doi.org/10.7554/eLife.29330.003

The following source data is available for figure 1:

**Source data 1.** The contents of hESC-CM differentiation media.
DOI: https://doi.org/10.7554/eLife.29330.004

of MitoTracker and MF20 expression even in the presence of 5 mM or 25 mM glucose (*Figure 2—figure supplement 1C*), suggesting that the effect is specific to glucose and not to changes in osmotic pressure. Consistently, flow cytometry showed a significant increase in cell size under glucose-restricted conditions (*Figure 2I*). Together, these results demonstrate that glucose dose-dependently suppresses the maturation of cardiomyocyte cellular architecture and the upregulation of cardiac genes in hESC-CMs.

## Glucose reduction promotes functional maturation of hESC-CMs

We next compared the metabolic and functional maturity of hESC-CMs cultured in the presence and absence of glucose by six methods. First, hESC-CMs were stained with JC-1, a green fluorescent dye that generates red fluorescence upon formation of aggregates in active mitochondria. The level of red fluorescence is often used as an indicator of mitochondrial membrane potential and, therefore, mitochondrial activity. Immunofluorescent staining revealed that mitochondria in glucose-reduced hESC-CMs are more elongated (*Figure 3A*), and flow cytometry analysis revealed that the level of JC-1 aggregation is significantly higher in glucose-reduced hESC-CMs cultured in both

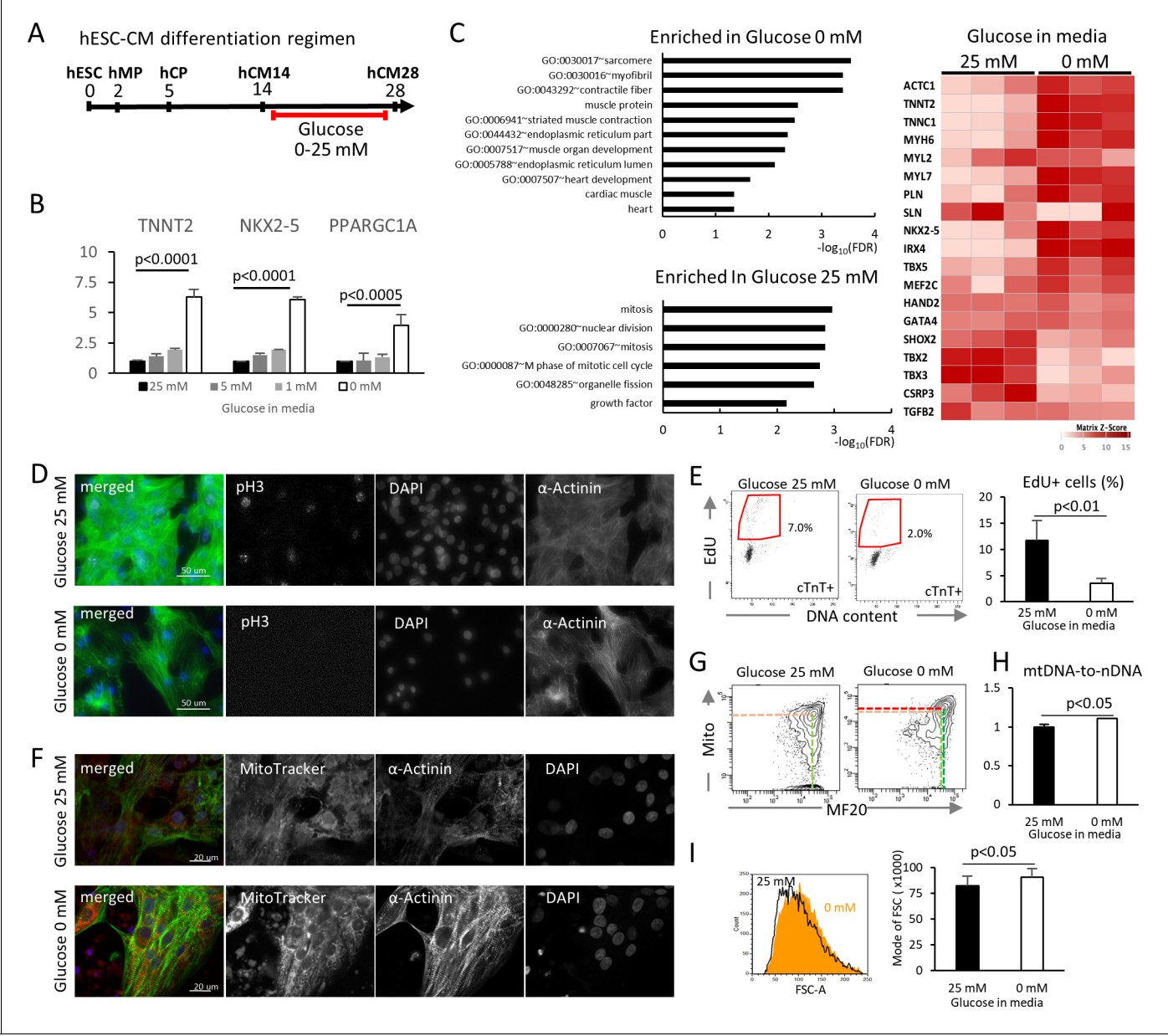

**Figure 2.** Glucose reduction promotes maturation of hESC-CMs. (**A**) Experimental regimen. hESC-CMs are differentiated in the medium containing 25 mM glucose until day 14, when ~ 90% of the cells are already MF20+. Cells are analyzed at day 28 unless otherwise specified. (**B**) Relative mRNA expression of *TNNT2*, *NKX2-5*, and *PPARGC1A* as determined by qPCR. All these markers are upregulated in hESC-CMs in glucose-deprived conditions (n = 3, mean ± SD, p-value by one-way ANOVA test). (**C**) Pathway analysis of differentially expressed genes in 0 mM glucose (top left panel), and of differentially expressed genes enriched in hESC-CMs in 25 mM glucose (bottom left) based on RNA-seq data. The heatmap (right panel) shows the relative expression of representative cardiac genes compared between hESC-CM cultured with 25 mM glucose or without glucose. (**D**) Assessment of mitotic activity by pH3 immunostaining. Representative images of three independent experiments. (**E**) Assessment of mitotic activity. Representative data from EdU flow cytometry (left) and quantitation of %EdU+ cardiomyocytes (right) are shown. (n = 3, mean ± SD, p<0.01 by t-test.) (**F**) Assessment of the maturity of cardiomyocytes by MitoTracker (mitochondrial content) and α-actinin staining. Representative images of three independent experiments. (**G**) Assessment of the maturity of cardiomyocytes by flow cytometry for MF20 and MitoTracker. Representative images of at least three independent experiments are shown. (**H**) Assessment of mitochondrial DNA contents obtained by quantitative PCR for mitochondrial and nuclear DNA (n = 4, mean ± SD, p<0.05 by t-test). (**I**) Assessment of the cell size by forward scatter (FSC) from flow cytometry data. At least 10,000 cells were measured for each sample. Representative histogram from three flow cytometry data for each group (left) and the geometrical means of FSC (right). (n = 3, mean ± SD, p<0.05 by t-test.)

DOI: https://doi.org/10.7554/eLife.29330.005

The following figure supplement is available for figure 2:

*Figure 2 continued on next page*

*Figure 2 continued*

**Figure supplement 1.** Glucose reduction promotes maturation of hESC-CMs.
DOI: https://doi.org/10.7554/eLife.29330.006

regular 0.9 mM and 10 mM pyruvate conditions. (*Figure 3B*). These data suggest that high glucose inhibits the functional maturation of mitochondria.

Second, intracellular lactate levels of hESC-CMs were measured using Laconic, a fluorescence resonance energy transfer (FRET)-based lactate nanosensor (*San Martín et al., 2013*). After glucose deprivation, the Laconic construct was introduced into hESC-CMs via adenovirus as described previously (*John et al., 2008*). In hESC-CMs differentiated in standard 25 mM glucose, bath-applied lactate (4 mM) caused a rapid increase in intracellular lactate (measured as a decrease FRET ratio), demonstrating the efficacy of the probe. Subsequent addition of pyruvate evoked a similar but smaller elevation of intracellular lactate. Under these conditions, inhibition of mitochondrial respiration with sodium cyanide (NaCN) had only a minor effect on the intracellular lactate level. This result suggests that hESC-CMs that are differentiated in standard 25 mM glucose do not actively metabolize pyruvate (*Figure 3C*, upper panel). By contrast, addition of 4 mM pyruvate to glucose-reduced hESC-CMs did not cause intracellular lactate accumulation, and addition of NaCN in the presence of pyruvate resulted in a substantial increase in lactate level. This result is consistent with pyruvate utilization by mitochondria (*Figure 3C*, lower panel). Together, these data suggest that mitochondria metabolize pyruvate in hESC-CMs that are cultured in glucose-reduced conditions, but not in the presence of glucose.

Third, we assessed cellular respiration of hESC-CM using the XF24 Extracellular Flux Analyzer (Seahorse Bioscience), in which oxygen consumption rate (OCR) was measured in real time in a basal state and in response to oligomycin (an ATP synthase inhibitor), FCCP (carbonilcyanide *p*-triflouromethoxyphenylhydrazone, a mitochondrial uncoupler), and rotenone or myxothiazol (complex I or III inhibitors, respectively) (*Figure 3D*). Although base-line mitochondrial respiration was not changed (*Figure 3E*), ATP-linked respiration was elevated in the no glucose condition (*Figure 3F and G*). Glucose-reduced hESC-CMs also demonstrated substantially larger maximum respiration capacity, as indicated by the response to FCCP (*Figure 3H*). These results corroborate the increased capacity of cellular respiration in hESC-CMs cultured under low glucose conditions.

Fourth, the $Ca^{2+}$ kinetics of hESC-CMs were assessed using a $Ca^{2+}$ transient assay (*Shimizu et al., 2015*). Although the peak amplitude of the transient (ΔF/F0) did not show a significant difference, the maximum upstroke ($V_{max}$) was significantly faster and the time to 50% decay was significantly shorter in the glucose-reduced group (*Figure 3I*). This pattern is consistent with the previous report demonstrating the role of thyroid hormone on hiPSC-CM maturation (*Yang et al., 2014*), and suggestive of an inhibitory role of high glucose on hESC-CM maturation.

Fifth, taking advantage of our monolayer culture system, we examined electrophysiological properties, using a multi-electrode array (MEA) culture plate as reported previously (*Zhu et al., 2017*). Maximum upstroke velocity ($dV/dt_{max}$) of field potential is a reliable parameter for approximating the electrophysiological maturity of cardiomyocytes derived from pluripotent stem cells (*Haase et al., 2009*; *Ma et al., 2011*; *Zhang et al., 2009*). Compared with hESC-CMs cultured in 25 mM glucose, those hESC-CMs cultured in the absence of glucose displayed a significant increase in $dV/dt_{max}$ (*Figure 3—figure supplement 1A–C*).

Finally, the monolayer culture method allowed us to measure cell contractility by digital image correlation using the MotionGUI program

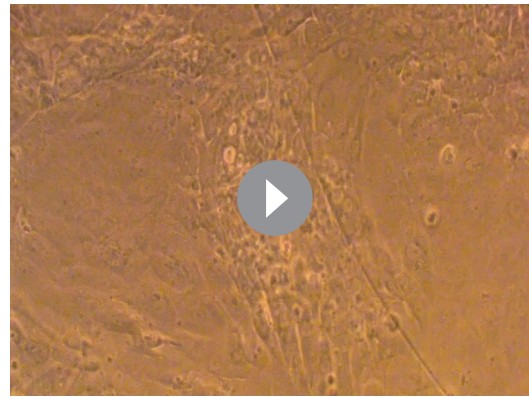

**Video 1.** Beating hESC-CMs differentiation in glucose 25 mM medium. This video shows beating of hESC-CMs differentiated in the medium containing 25 mM glucose from day 14 for 7 days.
DOI: https://doi.org/10.7554/eLife.29330.019

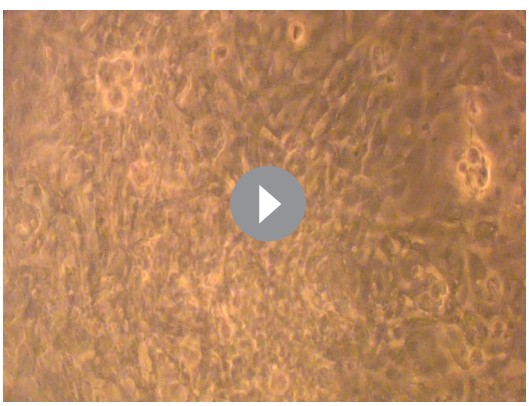

**Video 2.** Beating hESC-CMs differentiation in glucose 0 mM medium. This video shows beating of hESC-CMs differentiated in the medium containing 0 mM glucose from day 14 for 7 days.
DOI: https://doi.org/10.7554/eLife.29330.020

(*Huebsch et al., 2015*). Both average maximum contraction speed and average maximum relaxation speed were higher in those hESC-CMs cultured in 0 mM glucose (*Figure 3J*). Together, these data suggest that glucose reduction promotes the functional maturation of hESC-CMs at the metabolic, electrophysiological, and biomechanical levels.

## Glucose blocks cardiac maturation through the pentose phosphate pathway

Having determined that glucose reduction induces cardiac maturation at morphological, genetic metabolic, and functional levels, we next sought to analyze the mechanism by which glucose blocks cardiac maturation. Glucose is metabolized by multiple pathways involving both catabolic reactions (anaerobic glycolysis and the aerobic TCA cycle) and anabolic reactions (PPP, the hexosamine pathway, etc.). We first examined the impact of glucose reduction on the global metabolomics signature. Mass spectrometry revealed that glucose deprivation resulted in a significant decrease in the levels of the metabolites in purine metabolism, pyrimidine metabolism, the PPP, the hexosamine pathway, and glycolysis, whereas lipid precursors, amino acids, glutamine, and glutamate, as well as urea cycle metabolites, were not significantly affected (*Figure 4A* and *Figure 4—figure supplement 1*). ATP levels were not significantly different under glucose-restricted conditions (*Figure 4B*), neither were any specific stress pathways significantly increased when examined by RNA-seq, suggesting that the cells were not energy-starved in the absence of glucose in the cell culture media.

To identify the metabolic pathway responsible for the improved cardiac maturation by glucose reduction, we conducted a systematic screening using chemical inhibitors for the various glucose metabolic pathways in the monolayer 384-well format (*Figure 4C*). Consistent with flow cytometry for MitoTracker and MF20 (*Figure 2—figure supplement 1C*), 2-DG dose-dependently abolished the glucose-dependent inhibition of *TNNT2* and *NKX2-5* mRNA levels in hESC-CM (*Figure 4D* and *Figure 4—figure supplement 4A*). 3PO (3-[3-pyridinyl]-1-[4-pyridinyl]-2-propen-1-one), an inhibitor of the phosphofructokinase PFKFB3 (which is a regulator of PFK1), did not affect the level of hESC-CM maturity at any concentration of glucose (*Figure 4E* and *Figure 4—figure supplement 4B*). The failure of 3PO to recapitulate the effect of glucose deprivation suggests that glucose metabolites downstream of PFK1 are not essential for the glucose-dependent inhibition of cardiac maturation. Consistently, sodium oxamate (a lactate dehydrogenase [LDH] inhibitor) did not block the inhibitory effect of glucose (*Figure 4F* and *Figure 4—figure supplement 4C*). Interestingly, however, 6AN (6-[cyclohexa-2,5-dien-1-ylideneamino] naphthalene-2-sulfonate) and DHEA (didehydroepiandrosterone), both inhibitors of glucose-6-phosphate dehydrogenase (G6PD) in the oxidative arm of the PPP, recapitulated glucose reduction (*Figure 4G and H*, *Figure 4—figure supplement 4D–E*). As summarized in *Figure 4—figure supplement 3*, our chemical inhibitor screening suggests that the PPP plays a critical role in the inhibition of cardiac maturation and that blocking this pathway by either use of chemical inhibitors or glucose deprivation induces cardiac maturation. Although mitochondria are a major source of reactive oxygen species (ROS) and physiological levels of ROS promote cellular differentiation (*Crespo et al., 2010*), the level of ROS measured by DCFDA (dichlorodihydrofluorescein diacetate) did not increase in the absence of glucose and neither did ROS inhibition had a significant impact on *TNNT2* expression level (*Figure 4—figure supplement 3F*), suggesting that the increase in ROS is not responsible for the induction of cardiac maturation.

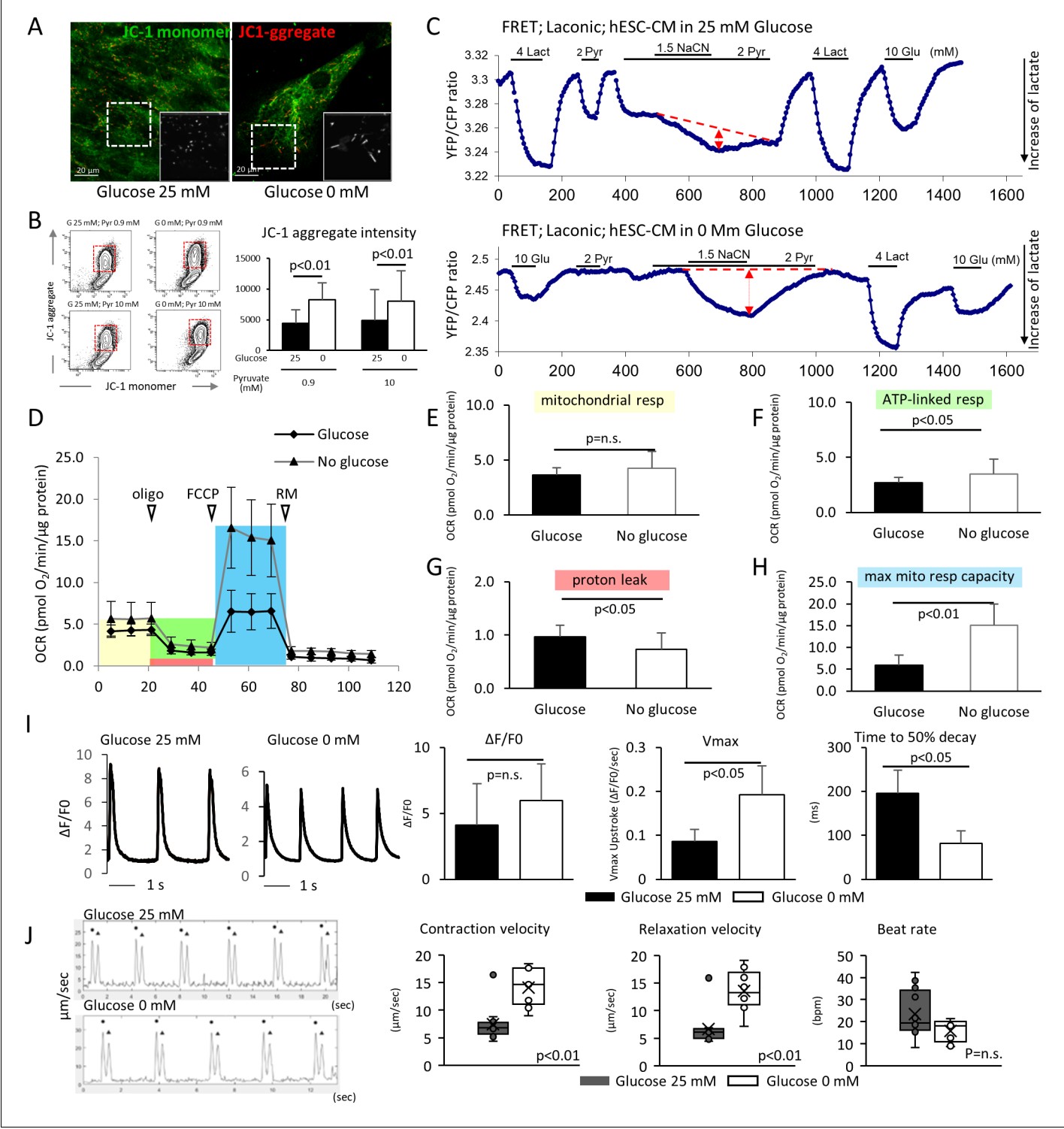

**Figure 3.** Glucose deprivation promotes the functional maturation of hESC-CM. (**A**) Representative images of mitochondrial membrane potential assay using JC-1 dye in hESC-CMs cultured in the presence (left) and absence (right) of glucose. Note the elongated mitochondria in the hESC-CMs cultured in 0 mM glucose. (**B**) Flow cytometry analyses of JC-1of hESC-CMs cultured in different concentrations of glucose and pyruvate. Representative flow cytometry profile (left) and the quantitation of the intensity of JC-1 aggregates (right). (Mean intensity ± SD, p<0.01 by one-way ANOVA test.) (**C**) Changes in intracellular lactate level measured as YFP/CFP ratio with Laconic, a fluorescence resonance energy transfer (FRET)-based biosensor of lactate, in hESC-CMs cultured in the presence (upper panel) or absence (lower panel) of glucose. Increase in intracellular lactate level is indicated by a downwards shift in the trace. Note that, in hESC-CMs cultured in the absence of glucose, the addition of pyruvate (Pyr) does not lead to an increase in

*Figure 3 continued on next page*

*Figure 3 continued*

intracellular lactate. Data are representative of two independent experiments. (**D–H**) Oxygen consumption rate (OCR) measured with a Seahorse analyzer (**D**). ESC-CMs cultured in the absence of glucose show comparable mitochondrial respiration (**E**), but greater ATP-linked respiration (**F**), andmaximum mitochondrial respiration capacity (**H**) with slight difference in proton leak (**G**), suggesting that glucose-deprivation potentiates OXPHOS. (n = 19, each; mean ± SD, p<0.01 by t-test.) (**I**) Calcium transient assay of hESC-CMs cultured in the presence or absence of glucose. Representative waves, Vmax, ΔF/F0, and time to 50% decay are shown. (n = 10 (G25) and 11 (G0), p values by t-test.) (**J**) Motion speed analysis by digital image correlation. The first (•) and the second peak (▲) of the duplex represent the contraction and the relaxation speed, respectively (left). Contraction velocity, relaxation velocity, and beat rate are shown. (n = 12 (G25) and 9 (G0); mean ± SD, p values by t-test.)
DOI: https://doi.org/10.7554/eLife.29330.007
The following figure supplement is available for figure 3:

**Figure supplement 1.** Electrophysiological analyses of the maturity of hESC-CMs by multi-electrode array (MEA).
DOI: https://doi.org/10.7554/eLife.29330.008

## Nucleotide metabolism regulates cardiomyocyte maturation

The oxidative arm of the PPP generates two major products: reducing power in the form of NADPH and 5-carbon sugars that supply the backbone for nucleotide biosynthesis. To test whether glucose level impacts cardiac maturation by affecting nucleotide biosynthesis, we rescued nucleotide synthesis by adding uridine to hESC-CMs cultured in low glucose media. Under glucose starvation, supplementation of uridine is known to rescue the growth of bacteria, yeast and malignant cells (*Linker et al., 1985*). In our hESC-CM culture system, uridine restored cell proliferation even in low-glucose conditions (*Figure 5A and B*). Interestingly, uridine dose-dependently reduced the level of *TNNT2* even in glucose-deprived conditions (*Figure 5C* and *Figure 5—figure supplement 1A*), suggesting that glucose-mediated inhibition of cardiac maturation is dependent on the supply of nucleotides and not of NADPH.

In order to test whether nucleotides are necessary for the glucose-dependent inhibition of cardiac maturation, nucleotide biosynthesis was blocked by multiple methods. Addition of an excess amount of thymidine (unlike uridine) blocks the synthesis of DNA by inhibiting the formation of deoxycytidine (i.e., the thymidine block method), which is commonly used to synchronize the cell cycle (*Reichard et al., 1960*; *Xeros, 1962*). When excess thymidine was added to hESC-CMs, the expression of *TNNT2* and *NKX2-5* were increased (*Figure 5D* and *Figure 5—figure supplement 1B*). To further confirm this effect, we blocked deoxynucleotide synthesis by hydroxyurea (HU), an inhibitor of ribonucleotide reductase (RNR) that catalyzes the formation of deoxyribo-nucleotides. Consistent with the results from thymidine block, HU dose-dependently induced the expression of *TNNT2* and *NKX2-5* (*Figure 5E* and *Figure 5—figure supplement 1C*). RNAi-based knockdown of RRM2B, a key subunit of RNR, also resulted in a significant increase in *TNNT2* expression level, even in the presence of 25 mM glucose (*Figure 4—figure supplement 2*). Together, these gain- and loss-of-function data suggest that nucleotide biosynthesis is a key regulatory pathway of the pro-mitotic–anti-maturation effect of glucose.

## Nucleotide deprivation, not cell cycle block, induces cardiomyocyte maturation

Nucleotide synthesis is a key step in DNA replication and thus cell cycling. Therefore, it is not clear whether the maturation of hESC-CMs that results from deprivation of glucose is due to the cell-cycle block in general or to the effects of nucleotides themselves. To examine whether cell-cycle arrest in general is an essential trigger for cardiac maturation, we blocked the mitotic activity of hESC-CMs by a CDK4/6 inhibitor and paclitaxel (Taxol; an inhibitor of microtubule breakdown), both of which block the cell cycle without directly inhibiting the nucleotide kinetics. Interestingly, neither CDK4/6 inhibitor nor paclitaxel induced cardiac maturation (*Figure 5—figure supplement 2A–B*). Together, these data suggest that cell-cycle arrest by itself is not crucial for the promotion of cardiac maturation. Rather, nucleotide deprivation is a key mechanism for cardiac maturation.

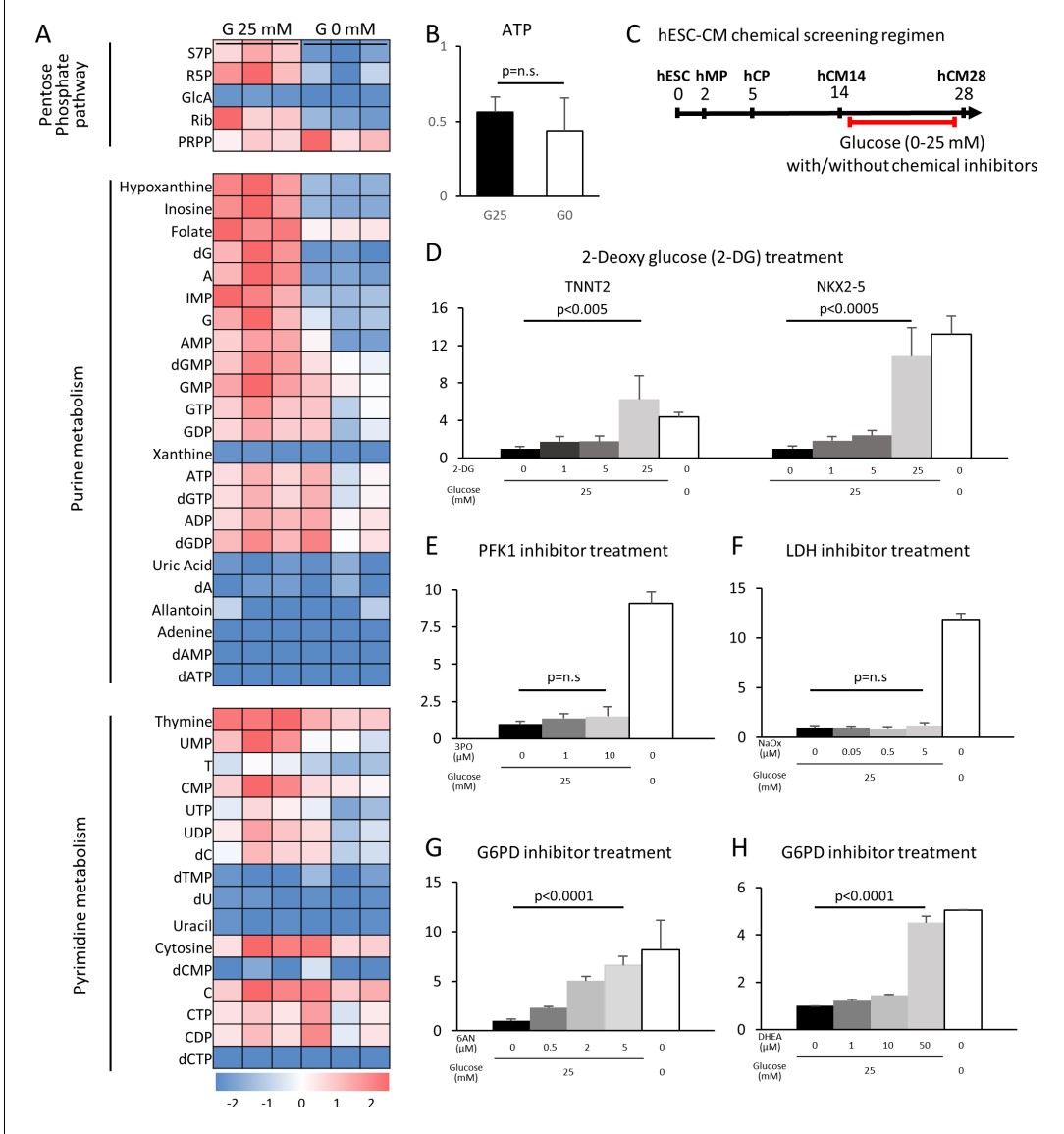

**Figure 4.** The pentose phosphate pathway inhibits cardiac maturation. (**A**) Heatmap presentation of the metabolomics analysis of hESC-CMs cultured in the presence or absence of glucose. Note the decrease in the metabolites in purine metabolism, pyrimidine metabolism and the pentose phosphate pathway (PPP) in glucose-deprived conditions (n = 3, each). GlcA, glucuronic acid; R5P, ribose 5-phosphate; Rib, ribose; PRPP, phosphoribosyl pyrophosphate; S7P, sedoheptulose-7-phosphate. See also *Figure 4—figure supplement 2*. (**B**) ATP levels of hESC-CMs in 25 mM glucose medium (G25) and glucose-depleted medium (G0) (n = 3, mean ± SD, p = n .s. by t-test.) (**C**) Experimental regimen for chemical inhibition of glucose metabolic pathways. hESC-CMs are cultured in the medium containing four different glucose levels and different chemical inhibitors. See also *Figure 4—figure supplement 3*. (**D**) Relative mRNA expression of *TNNT2* and *NKX2-5* in different concentrations of glucose and 2-DG, a competitive inhibitor of glucose. 2-DG restored cardiac maturation in the presence of glucose. (**E–H**) Relative mRNA expression of *TNNT2* in 0–25 mM glucose and with different chemical inhibitors of the glucose metabolic pathways: 0–10 μM 3PO (3-(3-pyridinyl)−1-(4-pyridinyl)−2-propen-1-one, a phosphofructokinase [PFK] inhibitor) (E); 0–5 mM sodium oxamate (NaOX; a lactate dehydrogenase [LDH] inhibitor) (F); 0–5 μM 6AN (6-aminonicotinamide, a glucose 6-phosphate dehydrogenase [G6PD] inhibitor) (G); and 0–50 μM DHEA (dehydroepiandrosterone, a G6PD inhibitor) (H). (n = 3, each. mean ± SD; p-value by one-way ANOVA.) See also *Figure 4—figure supplement 4*.

DOI: https://doi.org/10.7554/eLife.29330.009

The following figure supplements are available for figure 4:

**Figure supplement 1.** Metabolomics analyses by mass spectrometry.
DOI: https://doi.org/10.7554/eLife.29330.010

**Figure supplement 2.** RNAi knockdown of glucose metabolic enzymes.
DOI: https://doi.org/10.7554/eLife.29330.011

*Figure 4 continued on next page*

*Figure 4 continued*

**Figure supplement 3.** The pentose phosphate pathway inhibits cardiac maturation.
DOI: https://doi.org/10.7554/eLife.29330.012
**Figure supplement 4.** Summary of the impact of glucose metabolism inhibitors on cardiac maturity.
DOI: https://doi.org/10.7554/eLife.29330.013

## Glucose uptake is progressively suppressed during physiological cardiogenesis in utero

These in vitro data suggest that glucose reduction promotes cardiac maturation while inhibiting cardiomyocyte proliferation. An intriguing possibility is that the same mechanism underlies cardiac maturation in the in vivo natural counterpart. During normal embryogenesis, however, the blood glucose level is primarily regulated by maternal metabolism and stays relatively stable in utero, leading us to hypothesize that cellular glucose uptake becomes restricted during late fetal stages. To test this possibility, we measured the glucose uptake in the fetal hearts using $^{18}$F-labeled 2-deoxy-2-fluoroglucose (FDG). $^{18}$F-labeled FDG was injected intravenously via the maternal tail vein at E10.5, E12.5, and E15.5 or intraperitoneally into P1 and P7 pups. After 2 hr, the mice were imaged by PET/CT, the hearts were dissected, and cardiac accumulation was measured quantitatively. Interestingly, the normalized cardiac accumulation progressively and rapidly decreased from E10.5 to P7, with 0.11% and 0.05% $^{18}$F-FDG uptake in P1 and P7 hearts, respectively, compared to E10.5 hearts (*Figure 6*). These data suggest that cardiac glucose uptake becomes significantly restricted during late gestational and early postnatal stages, creating an intracellular glucose deprivation condition during natural in vivo development.

## Hyperglycemia promotes the proliferation and inhibits the maturation of cardiomyocytes in utero

We next tested whether hyperglycemia promotes the proliferation and inhibits the maturation of cardiomyocytes in vivo using fetuses and neonates from diabetic pregnancy. Akita heterozygous mice carry a single amino-acid substitution in the *Ins2* gene and exhibit multiple disorders associated with maturity-onset diabetes of the young (MODY) (*Barber et al., 2005*; *Fujita et al., 2001*; *Wang et al., 1999*; *Yaguchi et al., 2003*; *Yoshioka et al., 1997*). By crossing an Akita female with a wild-type male, we created a diabetic pregnancy condition in which wild-type fetuses (half of the pups in the litters) and their littermates are exposed to hyperglycemia (*Figure 7A*). In the C57BL/6 background, the average blood-glucose levels of the Akita mothers that we used were significantly higher than those of sex-matched control littermates (215 ± 84 vs 71 ± 12 mg/dl, respectively; p<0.0005). Fetal and neonatal Tnnt2$^+$ cardiomyocytes from wild-type hearts from wild-type mothers and wild-type hearts from Akita mothers were examined for mitotic activity in an in vivo EdU incorporation assay at E16.5 and P0 stages, when cardiomyocytes are not yet multinucleated or multiploidic. As shown in *Figure 7B and C*, the number of cardiomyocytes in S phase was significantly higher at both E16.5 and P0 in the diabetes group. Histological analyses showed that the number of phosphorylated histone H3-positive cardiomyocytes (pH3$^+$/Tnnt2$^+$) is higher at E16.5 in the embryos from diabetic pregnancies (*Figure 7D and E*). These data suggest that fetal cardiomyocytes are more mitotic when exposed to maternal hyperglycemia.

To examine whether hyperglycemia inhibits the maturation of fetal cardiomyocytes in vivo, we analyzed the fetal and neonatal hearts from diabetic pregnancies. The level of *Tnnt2* expression was significantly lower in the hearts from the diabetic pregnancies (*Figure 7F*). A hallmark of the congenital heart disease associated with diabetic pregnancy is asymmetric cardiac hypertrophy. Although heart weight/body weight ratio did not show a difference in our mouse model, the thickness of left and right ventricular free walls was significantly increased in the hearts from diabetic pregnancies at P1 (*Figure 7G*). Consistently, the cardiomyocyte size measured by flow cytometry was significantly smaller in the diabetic pregnancy group (*Figure 7H*). These data suggest that overproliferation and/or delayed maturation underlie the pathological mechanism of cardiomyopathy associated with diabetic pregnancy.

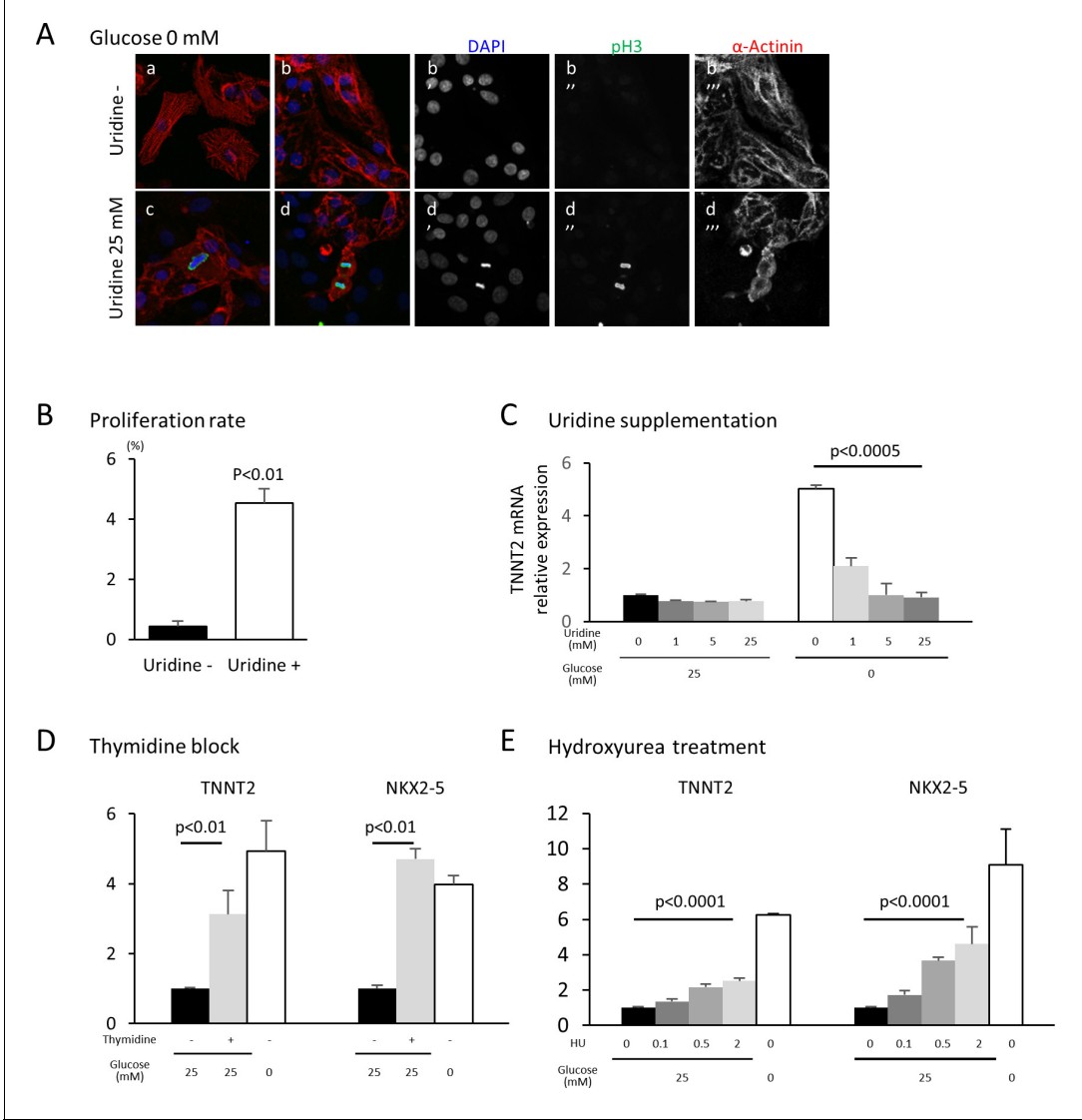

**Figure 5.** Nucleotide metabolism regulates cardiomyocyte maturation. (**A**) Glucose-deprived hESC-CMs are cultured in the absence (**a, b**) or presence (**c, d**) of 25 mM uridine, and stained for pH3 (a mitosis marker). The addition of uridine restored proliferative activity even in the absence of glucose. Data are representative of three independent experiments. (**B**) Proliferation rate, presented as number of pH3[+] cells/α-actinin[+]cells, seen in the stained images of *Figure 5A*. (n = 3, mean ± SD, p<0.01 by t-test.) (**C**) Relative mRNA expression of *TNNT2* in hESC-CMs in 25 mM or 0 mM glucose with 0 or 25 mM uridine. Uridine dose-dependently inhibited the *TNNT2* expression level in glucose-deprived conditions. (n = 3, mean ± SD, p<0.0005 by one-way ANOVA test.) See also *Figure 5—figure supplement 1A*. (**D**) Relative mRNA expression of *TNNT2* and *NKX2-5* in hESC-CMs cultured in 0–25 mM of glucose in the presence or absence of thymidine. Thymidine block increases the levels of *TNNT2* and *NKX2-5* (n = 3, mean ±SD, p<0.01 for a t-test comparing samples with or without 25 mM thymidine in 25 mM glucose. See also *Figure 5—figure supplement 1B*. (**E**) Relative mRNA expression of *TNNT2* and *NKX2-5* in hESC-CMs cultured in 0–25 mM glucose and 0–2 mM hydroxyurea (HU, a ribonucleotide reductase inhibitor). HU dose-dependently induced the expression of *TNNT2* and *NKX2-5* at 1, 5, and 25 mM glucose. (n = 3, mean ± SD, p-value by one-way ANOVA test.) See also *Figure 5—figure supplement 1C*.

DOI: https://doi.org/10.7554/eLife.29330.014

The following figure supplements are available for figure 5:

**Figure supplement 1.** Nucleotide inhibits hESC-CM maturation.
DOI: https://doi.org/10.7554/eLife.29330.015

**Figure supplement 2.** Nucleotide deprivation, not cell cycle block, is the primary inducer of cardiomyocyte maturation.
DOI: https://doi.org/10.7554/eLife.29330.016

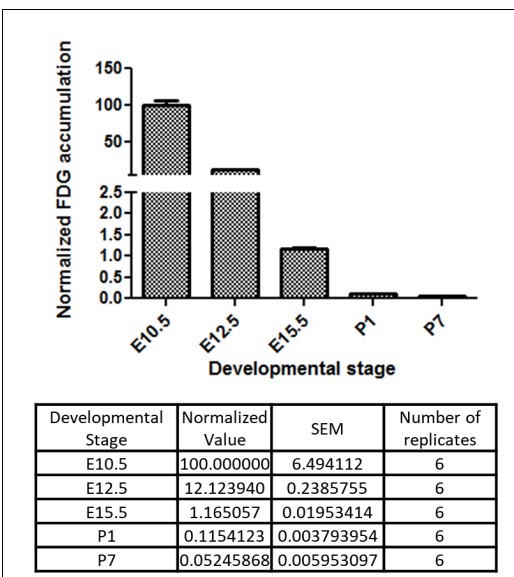

**Figure 6.** Developmental time course of cardiac glucose uptake measured by $^{18}$F-FDG accumulation. The radioactivity of the entire heart was measured by γ-counter after tail vein i.v. (fetus) or i.p. (neonates) injections. Values were normalized to heart weight and total body signal (heart values/[heart weight * total body values]). Note a drastic decrease in glucose uptake during late gestational and neonatal stages (n = 6, each; p<0.0001 by one-way ANOVA test.)
DOI: https://doi.org/10.7554/eLife.29330.017

| Developmental Stage | Normalized Value | SEM | Number of replicates |
|---|---|---|---|
| E10.5 | 100.000000 | 6.494112 | 6 |
| E12.5 | 12.123940 | 0.2385755 | 6 |
| E15.5 | 1.165057 | 0.01953414 | 6 |
| P1 | 0.1154123 | 0.003793954 | 6 |
| P7 | 0.05245868 | 0.005953097 | 6 |

## Discussion

In summary, we investigated the role of glucose in cardiac formation, and discovered (1) that glucose dose-dependently inhibits cardiac maturation in hESC-CMs, (2) that the maturation-inhibitory effect is dependent on nucleotide biosynthesis via the PPP, (3) that the developing heart accomplishes intracellular glucose starvation by limiting glucose uptake in late gestational stages during normal embryogenesis, and (4) that perturbation of the environmental glucose level in diabetic pregnancy affects natural cardiomyocyte maturation in vivo.

Cardiomyocytes switch their main energy substrate from glucose (or other carbohydrates) to fatty acids shortly after birth. This metabolic switch has long been believed to be an adaptation of cardiomyocytes to facilitate more efficient production of ATP. However, our study has revealed that a drastic suppression of glucose uptake occurs during gestational stages, long before the metabolic switch after birth (*Figure 5*). Our in vitro hESC-CM glucose-deprivation experiments mimic this in vivo glucose starvation. The results suggest that glucose deprivation induces cardiac maturation at genetic, morphological, metabolic, electrophysiological and biomechanical levels (*Figures 2* and *3*), suggesting that glucose is a negative regulator of the maturation of fetal cardiomyocytes in vitro and in vivo, as well as a positive regulator of the mitotic activity of these cells. Perturbation of the natural glucose starvation results in higher mitotic activity and lower maturity of cardiomyocytes in vivo (*Figure 7*). Together, our results suggest that the metabolic switch during perinatal stages is necessary not only to meet the energy demand but also to induce the genetic program that facilitates the maturation of the cardiomyocytes in vivo. An important question that is yet to be answered is how the drastic suppression of intracellular glucose uptake is achieved in the fetal heart. As the fetal glucose environment is primarily determined by maternal metabolism and kept relatively constant in utero, one possibility is that the glucose uptake is limited at the glucose transporter level in fetal cardiomyocytes. In fact, fetal heart switches its glucose transporter isoform at around this stage. Understanding how glucose regulates the genetic program and how the glucose uptake is regulated at the genetic level will be key to further dissecting the cross-talk between the genetic and non-genetic factors governing heart formation.

### Nucleotide biosynthesis via the PPP as a key mechanism in balancing proliferation and maturation

Glucose is the most fundamental and commonly available nutrient for the cells. Hence, the activity of the glucose metabolic pathways is tightly regulated in cells. Glucose is broken down to extract energy through the glycolysis pathway and also shunts to supply 5-carbon sugars and NADPH through the PPP. In our study, chemical inhibition of glucose metabolic pathways in hESC-CMs revealed that it is not the catabolic breakdown of glucose to extract energy but rather the anabolic use of glucose to build nucleotides that is responsible for the glucose-dependent inhibition of cardiac maturation. Most of the proliferating cells synthesize nucleotides de novo from glucose, glutamine, and $CO_2$. In our hESC-CM experiments, blocking the PPP and nucleotide biosynthesis inhibited the glucose-mediated induction of mitosis and suppression of maturation, and

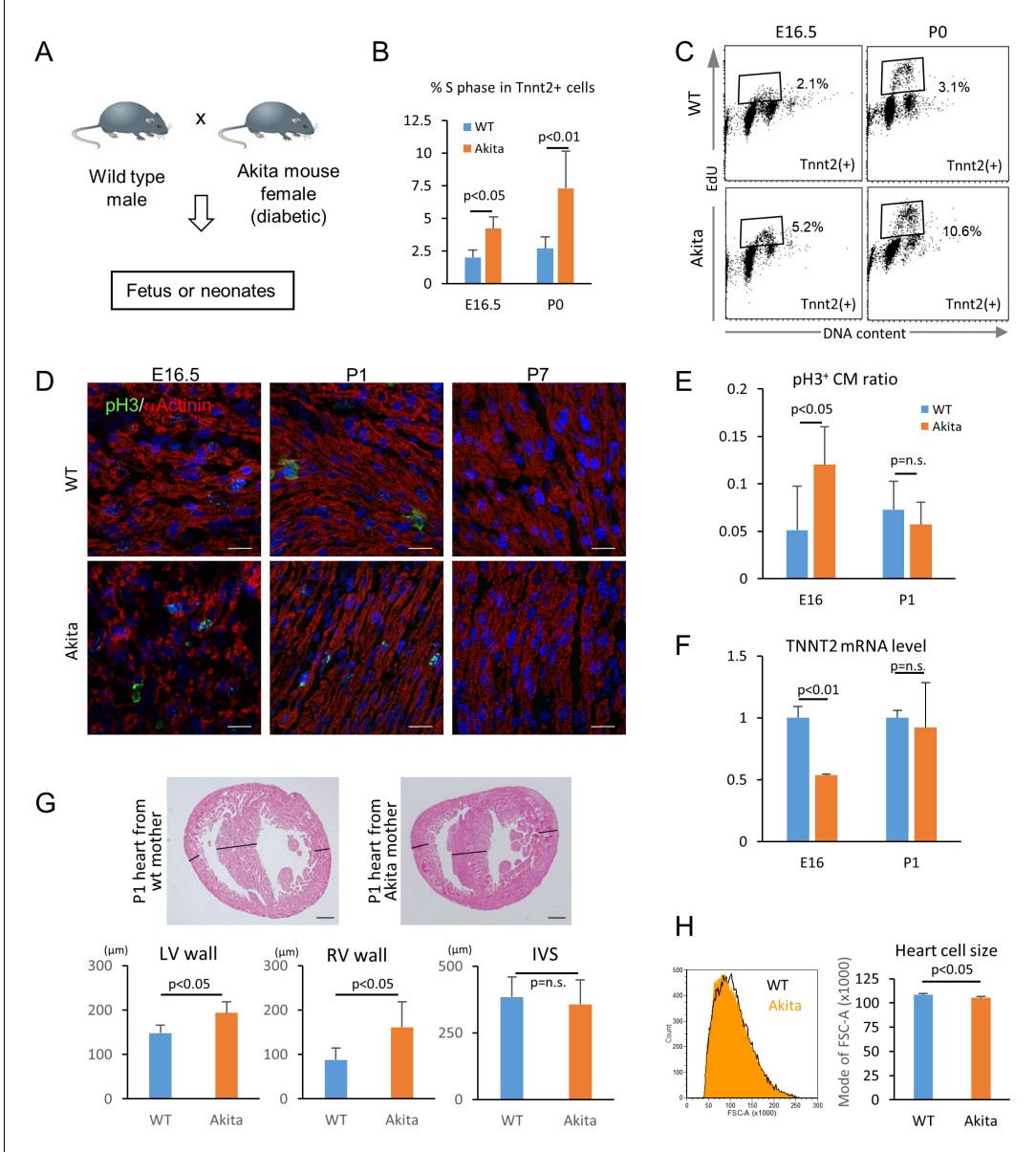

**Figure 7.** Hyperglycemia promotes the proliferation and inhibits the maturation of fetal cardiomyocytes in utero. (**A**) Diagram illustrating in vivo analysis of the impact of maternal hyperglycemia on fetal heart development using a diabetic mouse model (Akita). (**B,C**) Cell cycle analyses of fetal and neonatal cardiomyocytes from normal and diabetic pregnancies. Tnnt2-positive cardiomyocytes from diabetic Akita mothers show a higher percentage of cells in S-phase at both E16.5 and P0. (n = 7, each; data shown are mean ± SD, p-value by t-test). (**D,E**) Double immunostaining for phospho-histone H3 (pH3, green) and α-actinin (red) of the heart from normal and diabetic pregnancies at E16.5, P1 and P7 (**D**). %pH3[+] cells within α-actinin[+] cardiomyocytes (CM) from normal (WT) and diabetic (Akita) pregnancies at E16.5 and P1 (**E**). At least 10,000 cardiomyocytes were counted for each of five hearts. (n = 7, each; mean ± SD, p-value by t-test.) (**F**) qPCR analysis for *Tnnt2* expression in hearts from normal and diabetic pregnancies. The expression level is normalized to control at each stage (n = 3, each; p-value by t-test.) (**G**) Histological analysis of P1 hearts from normal (WT) and diabetic (Akita) pregnancies. Ventricular wall thickness (RV, right ventricle; LV, left ventricle) and interventricular septum (IVS) thickness were analyzed histologically. Scale bar = 200 μm. (n = 5 and 4 for WT and Akita, respectively; data shown are mean ± SD, p-value by t-test). (**H**) Cell-size analysis of the cells isolated from P1 hearts from normal (WT) and diabetic (Akita) pregnancies using forward scatter by flow cytometry (FSC). At least 25,000 cells were measured per sample. Representative histogram from three flow cytometry measurements for each group (left) and the geometrical means of FSC (right). (n = 3, p<0.05 by t-test).

DOI: https://doi.org/10.7554/eLife.29330.018

supplementation of nucleotides was sufficient to recapitulate the effect of glucose (*Figures 4* and *5*). These data suggest that nucleotide biosynthesis via the PPP is the key regulator of the pro-mitotic/anti-maturation effect of glucose. Interestingly, cardiomyocyte maturation was not fully induced by blocking of the cell cycle with a CDK4/6 inhibitor or paclitaxel, neither of which directly impact the nucleotide kinetics. Therefore, it is not the cell cycle in general but the nucleotides themselves that blocks the maturation (*Figure 5—figure supplement 2*). It is well-documented that there is generally an inverse correlation between cell proliferation and differentiation during developmental stages (*Ruijtenberg and van den Heuvel, 2016*). Our data raise an intriguing possibility that nucleotide biosynthesis serves as a nodal point balancing cell proliferation and differentiation during development.

### Hyperglycemia as a potential teratogen in the fetal heart

Clinically, maternal diabetes can accompany multiple complications including neuropathy, microvasculopathy, nephropathy, and insulin resistance. Although meta-analysis predicts that hyperglycemia itself is a major teratogen during diabetic pregnancy (*Reece et al., 1996*), it is often difficult to dissect the impacts of maternal complications on CHD as they are often subclinical. To our knowledge, our in vitro study is the first to demonstrate that environmental glucose itself, if excessive, directly impacts cardiac differentiation. The formation of the heart is regulated by both genetic and non-genetic factors, with the latter playing important roles particularly during late-stage cardiogenesis. An interesting aspect of the interaction between genetic and non-genetic mechanisms is that they seem to reinforce each other mutually. Our data suggest that the glucose metabolic environment is, on the one hand, a consequence of changes in cardiac genetic program, and on the other hand, a cause of the changes in cardiac gene expression.

### Potential application to the translation

Understanding the metabolic signature of hESC-CMs will potentially open new methods for purifying these cells (*Tohyama et al., 2016*; *Tohyama et al., 2013*) or inducing their maturation (*Drawnel et al., 2014*). Considering that the inhibitors of the PPP and nucleotide biosynthesis have entered clinical trials for cancer treatment (*Tennant et al., 2010*; *Vander Heiden, 2011*), our data raise the possibility that manipulating this pathway may allow us to control the proliferation and maturation of cardiomyocytes for regenerative medicine.

With the advances in fetal diagnosis and surgical techniques, the number of CHD patients who survive childhood (and so have adult CHD) is growing rapidly by nearly 5% per year (*Brickner et al., 2000*). Maternal hyperglycemia is a common medical condition associated with 2–5-fold increase in CHD (*Centers for Disease Control, 1990*; *Simeone et al., 2015*; *Yogev and Visser, 2009*). Currently, 60 million women of reproductive age (18–44 years old) worldwide and approximately 3 million in the U.S. have diabetes mellitus. This number is predicted to double by 2030, posing a huge medical and economic burden (*Gabbay-Benziv et al., 2015*). Our findings will lay a foundation for understanding how the glucose environment regulates cardiogenesis and how disturbance of non-genetic factors affects the genetic program during the pathological development of the heart.

## Materials and methods

### Mouse and cell lines

Wild-type and Akita mice were maintained on the C57BL/6 background according to the Guide for the Care and Use of Laboratory Animals published by the US National Institute of Health (NIH Publication No. 85–23, revised 1996). Housing and experiments were performed according to the Institutional Approval for Appropriate Care and Use of Laboratory Animals by the UCLA Institutional Animal Care and Use Committee (Protocol #2008-127-07). H9 (WA09) and UCLA4 (UCLA stem cell core) hESC lines were maintained as described before (*Arshi et al., 2013*). Authentication of hESCs was achieved by confirming the expression of pluripotency genes and protein markers. hESCs were routinely verified as mycoplasma-free using a PCR-based assay. hESCs were grown and differentiated in a chemically defined condition (*Minami et al., 2012*; *Young et al., 2016*; *Zhu et al., 2017*). Usage of all the human embryonic stem cell lines is approved by the UCLA Embryonic Stem Cell

Research Oversight (ESCRO) Committee and the Institutional Review Boards (IRB) (approval #2009-006-04).

## RNA-seq and data analyses

For the RNA-seq analyses shown in *Figures 1D, E* and *2C*, RNA was extracted from hESC, hMP, hCP, hCM14, hCM28, hCM28 (in 25 mM glucose) and hCM28 (in 0 mM glucose) using TRIZOL (TheroFisher) and RNeasy kit (QIAGEN). 500 ng of DNaseI-treated RNA was used as input material for library preparation using the Illumina TruSeq mRNA kit (Illumina, RS-122–2001), according to manufacturer's instructions. Final libraries were sequenced as single-end 50 bp on the Illumina HiSeq2000 platform (GSE84814). Libraries for RNA-Seq shown in *Figures 1D, E* and *2C* were prepared with the KAPA Stranded RNA-Seq Kit. The workflow consists of mRNA enrichment, cDNA generation, end repair, A-tailing, adaptor ligation, strand selection and PCR amplification. Different adaptors were used for multiplexing samples in one lane. Sequencing was performed on an Illumina HiSeq 3000 for a paired end 2 × 150 run (GSE84815). A data quality check was done on an Illumina SAV. De-multiplexing was performed with the Illumina Bcl2fastq2 v 2.17 program. Short read sequences generated from an Illumina Sequencer were aligned to the UCSC human reference genome hg19 downloaded from support.Illumina.com (http://support.illumina.com/sequencing/sequencing_software/igenome.html) using TopHat from the Tuxedo Tools. The average overall read mapping rate reached over 82 percent (average = 82.43 percent). The output was in the form of BAM (Binary Sequence Alignment/Map format) files. These outputs contain information for assigning location and quantifying the short-read alignments obtained from the RNA-seq samples. This is necessary for downstream analyses such as annotation, transcript abundance comparison and polymorphism detection. Counts or gene expression matrices were generated using HTSeq, which quantified the reads per transcript. The expression matrices were log transformed and normalized using the rlog function in the Deseq2 package. The normalized gene expression matrices were used as input for SaVanT (Signature Visualization Tools), which allowed for the visualization of molecular signatures directly related to heart development as seen in *Figures 1D, E* and *2C*.

## Flow cytometry

hESC-CMs and mouse embryonic hearts were washed three times with PBS and incubated at 37°C in a dissociation enzyme solution with occasional pipetting to a single-cell suspension. The enzyme solution contained 1% penicillin/streptomycin (ThermoFisher, 15140–122), 10% fetal bovine serum (Hyclone), collagenase 2 mg/ml (Worthington, CLS-2), dispase 0.25 mg/ml (Gibco, 17105–041), and DNAase I (ThermoFisher) in PBS. The cells were analyzed with the following antibodies: MF20 (mouse, 1:100, Hybridoma Bank), Tnnt2 (rabbit, 1:250, Sigma-Aldrich), MitoTracker Orange CMTMRos (ThermoFisher), JC-1 (Abcam), and EdU (100 µl of 10 mM EdU solution per 10 g of mouse, ThermoFisher). For MF20 and Tnnt2, FITC-conjugated anti-mouse IgG secondary antibody (BD Biosciences) was used. Stained cells were analyzed by a flow cytometer (LSRII, BD Biosciences). Data analysis was performed using FACSDiva (BD Biosciences).

## Immunocytochemistry and morphological analysis

Cells were fixed with 4% paraformaldehyde, blocked for an hour with 5% normal goat serum, and incubated with mouse alpha actinin antibody (Sigma) followed by Alexa fluor 488-conjugated secondary antibody (ThermoFisher). Images were taken with Zeiss LSM780 confocal microscopy. Sarcomere lengths were analyzed using Zeiss Zen software.

## mtDNA-to-nDNA ratio analysis

Total DNA including mtDNA was extracted from cells using the PureLink DNA kit (ThermoFisher), and DNA purity and quantity were determined using a spectrophotometer. To determine the ratio between mitochondrial and nuclear DNA, qRT-PCR was performed on a Roche Lightcycler 480 using SYBR Green dye. Mitochondrial gene expression was corrected for nuclear gene expression values, and normalized to the value of the control group for each experiment as described before. Forward and reverse primer sequences are as follows: UUR forward, CAC CCA AGA ACA GGG TTT GT; UUR reverse, TGG CCA TGG GTA TGT TGT TA for mt DNA; B2-microglobulin forward, TGC TGT CTC CAT GTT TGA TGT ATC T; and B2-microglobulin reverse, TCT CTG CTC CCC ACC TCT AAG T.

## Ca$^{2+}$ transient assay

Ca$^{2+}$ transient was measured as described (*Shimizu et al., 2015*). Briefly, hESC-CMs cultured in the presence or absence of glucose were loaded with 5 μM fluo-4 AM and imaged in Tyrode buffer containing 138.2 mM NaCl, 4.6 mM KCl, 1.2 mM MgCl, 15 mM glucose and 20 mM HEPES according to the manufacturer's instruction. Images were recorded on a Zeiss LSM 780 confocal microscope. Data analysis was carried out using the Zeiss Zen and ImageJ.

## In vitro contractility assay

Contractility assessments were performed by utilizing a video-based technique with the UCSF Gladstone-developed Matlab program MotionGUI (*Huebsch et al., 2015*). The videos were converted from .mts to .avi format at native resolution using commercially available software and loaded into the MotionGUI program. The conversion between pixels and real distance was performed within the MotionGUI program using a reference image with unit divisions of 100 um, taken under the same objective and video zoom settings as the cell videos, to yield a pixel size of 0.681125. This pixel size was used for all contractility assessments. Motion vectors were calculated and the data were evaluated upon completion. All samples were subjected to the same post-processing procedures in order to ensure consistency during comparative analysis. Each video sample was post-processed using neighbor-based cleaning with the vector-based cleaning criterion within the program. The threshold for this post-processing method was set to two for all samples and was adequate for improving the signal-to-noise ratio enough to identify peaks clearly corresponding to beating events in most samples. A small number of videos suffering from significant noise issues were separately subjected to fast Fourier transform (FFT) frequency domain cleaning with a cut-off frequency of 1 Hz. Only one post-processing method was applied to one video at one time. All other parameters of the MotionGUI program not outlined here were set to their respective default values.

## Measurement of intracellular lactate level by Laconic

The lactate biosensor Laconic was a gift from Dr. Barros (*San Martín et al., 2013*). Overexpression of Laconic in hESC-CM was achieved using engineered adenoviruses encoding the construct. Expression of the construct was sufficiently high after 36–48 hr for FRET experiments or microscopy imaging. All cells were imaged live without fixation. Images (16 bit) were acquired using a microscope (Eclipse TE300; Nikon) fitted with a 60× (1.4 NA) oil immersion lens (Nikon) and equipped with a filter cube comprising a CFP bandpass excitation filter, 436/20b, together with a longpass dichroic mirror (455DCLP; Chroma Technology Corp). Two LEDs (Philips Lumileds), one emitting at 455 ± 20 nm (royal blue) and the other emitting at 505 ± 15 nm (cyan) were used as light sources. Ratiometric FRET measurements were obtained from the YFP and CFP images acquired simultaneously using a Dual View image splitter (Optical Insights) equipped with a 505 nm longpass dichroic filter to separate the CFP and YFP signals, a CFP emission filter (480/30), and a YFP emission filter (535/40) (*John et al., 2008*). Images were captured with a Cascade 512B digital camera (Photometrics). Reagents indicated in *Figure 3C* were added and followed by washing.

## XF24 extracellular flux analyzer

hESC-CMs were seeded onto a matrigel-coated XF24 Cell Culture Microplate (Seahorse Bioscience) at 2–7.5 10$^4$ cells/well with or without glucose (25 mM glucose of cardiac differentiation media). Oxygen consumption rate (OCR) was measured using an XF24 Extracellular Flux Analyser (Seahorse Bioscience) in unbuffered DMEM assay medium supplemented with 1 mM pyruvate, 2 mM glutamine and with or without 25 mM glucose. OCR was measured before and after the sequential addition of 0.75 μM oligomycin, 0.5 μM FCCP and 0.75 μM of rotenone/myxothiazol. OCR was normalized to protein concentration using a Bradford assay (Bio-Rad). Mitochondrial respiration was calculated as the difference between total and rotenone/myxothiazol rates. Maximal respiration was the response to FCCP. ATP-linked respiration was represented by the oligomycin-sensitive respiration rate, whereas uncoupled respiration was represented by the difference between oligomycin and rotenone/myxothiazol rates.

## Multi-electrode array

hESC-CMs at the stage of hCM14 were plated on microelectrode arrays (MEAs) containing 120 integrated TiN electrodes (30 μm diameter, 200 μm interelectrode spacing). The MEAs were placed in an incubator with a temperature of 37°C and 5% $CO_2$. Two days were given to allow the cardiomyocytes to well attach the MEAs before recording started. Local field potentials at each electrode were collected over a period of 5 min every day in total with a sampling rate of 1 KHz using the MEA2100-HS120 system (Multichannel systems, Reutlingen, Germany). Data analysis was carried out using the MC_DataTool (Multichannel Systems), Origin (OriginLab Corporation) and Matlab (Math-Works). Data shown are based on three independent hESC-CM preparations.

## Mass spectrometry-based metabolic measurements

The experiments were performed as described (*Krall et al., 2016*). Briefly, cells were seeded in 6-well plates, so that the final cell count at the time of metabolite extraction was about $7*10^5$ and this was even across all cell lines. To extract intracellular metabolites, cells were briefly rinsed with cold 150 mM ammonium acetate (pH 7.3), followed by the addition of 1 ml cold 80% MeOH on dry ice. Cell scrapers were used to detach cells, and the cell suspension was transferred into Eppendorf tubes. Extracted metabolites were transferred into glass vials and dried down under vacuum. For the LC-MS-based analysis, the samples were resuspended in 70% acetonitrile and 50 μl were injected onto a Luna NH2 column (150 mm x 2 mm, Phenomenex). Separation was achieved using A) 5 mM $NH_4AcO$ (pH 9.9) and B) ACN. The gradient started with 15% A) going to 90% A) over 18 min, followed by an isocratic step for 9 min and reversal to the initial 15% A) for 7 min. Metabolites were quantified with TraceFinder 3.3 using accurate mass measurements ($\leq$3 ppm) and retention times of pure standards. Data analysis was performed using the statistical language R.

## Gene expression analysis by quantitative reverse-transcriptase PCR

RNA was extracted from the tissue or the cells cultured with a specific concentration of glucose together with or without titrated metabolic pathway inhibitors using the Direct-zol RNA mini prep kit (Zymo research). RNA was reverse-transcribed into complementary DNA using the qScript cDNA synthesis kit (Quanta Biosciences). Quantitative reverse-transcriptase PCR was performed using Viia7 (Applied Biosystems/ThermoFisher). In *Figures 1B*, *2B* and *4C–E*, *Figure 4—figure supplement 3A–E*, *Figure 5—figure supplement 1A–C*, and *Figure 5—figure supplement 2A–B*, each bar represents the average of biological duplicates with at least three independent wells, each of which is triplicated for qPCR reaction. The relative mRNA level is normalized to the expression level of 25 mM glucose without any chemicals. Forward and reverse primer sequences are as follows: GAPDH forward, TTGAGGTCAATGAAGGGGTC; GAPDH reverse, GAAGGTGAAGGTCGGAGTCA; TNNT2 forward, CAGAGCGGAAAAGTGGGAAGA; TNNT2 reverse, TCGTTGATCCTGTTTCGGAGA; NKX2-5 forward, GTTGTCCGCCTCTGTCTTCT;NKX2-5 reverse, TCTATCCACGTGCCTACAGC; PPARGC1A forward, GGTGCCTTCAGTTCACTCTCA; and PPARGC1A reverse, AACCAGAGCAG-CACACTCGAT.

## $^{18}$F-FDG measurement by γ-counter

$^{18}$F-FDG was obtained from the UCLA Department of Nuclear Medicine. Warmed pregnant mice or pups were injected intravenously or intraperitoneally, respectively, with ~90 microCi (~3.33 MBq) of $^{18}$F-FDG. After 2 hr, the mice were sacrificed. Preliminary experiments suggested that 2 hr was sufficiently long for $^{18}$F-FDG to reach maximum accumulation in each organ and embryo. Fetal or neonatal hearts were separated from the other tissue (carcass), and the mass and radioactivity in both the hearts and the carcasses were measured using a standard balance and a Wizard 3' automatic gamma counter (Perkin Elmer), respectively. The radioactivity levels in the pup carcasses were higher than the detection limit of the gamma counter, so the expected gamma counter values for the pup carcasses were calculated on the basis of the decay-corrected injected dose of $^{18}$F-FDG and known conversion values between microCi and CPM on the gamma counter. To calculate the 'Normalized FDG accumulation', radioactive accumulation in each heart was divided by heart weight and then further divided by the total radioactivity in each embryo or pup. This last normalization is to account for differences in $^{18}$F-FDG injected dose and accessibility to the embryos and pups. Averages and

standard errors of the mean (SEM) were calculated, and the values were normalized such that E10.5 embryo FDG accumulation was set to 100.

## Immunostaining

Mouse embryos were isolated in cold PBS and fixed in 4% PFA for 1~2 hr, followed by equilibration in 30% sucrose in PBS solution overnight. The tissues were placed in 1:1 30% sucrose/OCT (Tissue-Tek, Electron Microscopy Sciences) solution for 1–2 hr, then in 100% OCT compound for 1 hr at 4°C, and embedded in 100% OCT compound, carefully oriented in Cryomolds (Ted Pella). The blocks were immediately frozen on dry ice with isopropanol and stored at −80°C. The sections were cut 5 µm with a Leica CM3050 S cryostat. The following primary and secondary antibodies were used: α-actinin (mouse, 1:200, Sigma-Aldrich), Phospho-Histone 3 (rabbit, 1:250, Millipore), as well as Alexa Fluor 488 (green)- and Alexa Fluor 594 (red)-conjugated secondary antibodies specific to the appropriate species, which were used (1:500; ThermoFisher) for fluorescent staining. Sections were mounted with antifade mounting medium with DAPI (ThermoFisher), and analyzed by using AxioImager D1 (Carl Zeiss Microimaging, Inc).

## Statistical analysis

ANOVA and Student's t-test were used to determine whether any statistically significant difference exists among independent groups.

## Acknowledgement

Authors thank Jinghua Tang and the BSCRC stem cell core, the BSCRC FACS core, and the metabolomics core for technical support. This work was supported by the Oppenheimer Foundation (AN), the Center for DMD at UCLA (AN), NIH NIAMS P30AR057230 (Musculoskeletal Core Center, AN), the National Center for Research Resources Grant S10RR026744 (KR), NIH DK094311 (AJL), NIH CA178415 (SKK), NIH HL126051 (AN), and NIH HL124503 (AZS, JKG, AN). XD is supported by a fellowship from the Chinese Scholarship Council of Chemistry and Chemical Engineering, Hunan University, Changsha, PR China.

## Additional information

### Competing interests

Norio Nakatsuji: Chief Advisor and shareholder of a stem cell-related start-up company, Stem Cell & Device Laboratory, Inc. Also a shareholder of ReproCELL, Inc. The other authors declare that no competing interests exist.

### Funding

| Funder | Grant reference number | Author |
| --- | --- | --- |
| Oppenheimer Foundation | | Atsushi Nakano |
| University of California, Los Angeles | Center for Duchenne Muscular Dystrophy, Pilot and Feasibility Seed Grant program | Atsushi Nakano |
| National Institute of Arthritis and Musculoskeletal and Skin Diseases | P30AR057230 | Atsushi Nakano |
| National Center for Research Resources | Grant S10RR026744 | Karen Reue |
| National Institutes of Health | DK094311 | Aldons J Lusis |
| National Institutes of Health | CA178415 | Siavash K Kurdistani |
| National Institutes of Health | HL126051 | Atsushi Nakano |

| National Institutes of Health | HL124503 | Adam Z Stieg |
| | | James K Gimzewski |
| | | Atsushi Nakano |
| Hunan University | Chinese Scholarship Council of Chemistry and Chemical Engineering | Xueqin Ding |

The funders had no role in study design, data collection and interpretation, or the decision to submit the work for publication.

## Author contributions

Haruko Nakano, Data curation, Formal analysis, Validation, Investigation, Visualization, Writing—original draft, Project administration, Writing—review and editing; Itsunari Minami, Xiuju Wu, Resources, Methodology; Daniel Braas, Data curation, Formal analysis, Visualization, Writing—review and editing; Herman Pappoe, Formal analysis, Visualization; Addelynn Sagadevan, Christopher Dunham, Data curation; Laurent Vergnes, Peter M Clark, Data curation, Formal analysis, Methodology, Writing—review and editing; Kai Fu, Data curation, Formal analysis, Validation, Visualization, Writing—review and editing; Marco Morselli, Formal analysis, Methodology; Xueqin Ding, Data curation, Formal analysis, Methodology; Adam Z Stieg, Matteo Pellegrini, Supervision, Funding acquisition, Writing—review and editing; James K Gimzewski, Karen Reue, Supervision, Funding acquisition; Aldons J Lusis, Resources, Supervision, Funding acquisition; Bernard Ribalet, Data curation, Formal analysis, Supervision, Investigation, Visualization, Methodology, Writing—review and editing; Siavash K Kurdistani, Heather Christofk, Supervision, Funding acquisition, Writing—original draft; Norio Nakatsuji, Supervision, Methodology; Atsushi Nakano, Conceptualization, Data curation, Formal analysis, Supervision, Funding acquisition, Investigation, Writing—original draft, Project administration, Writing—review and editing

## Author ORCIDs

Haruko Nakano  https://orcid.org/0000-0001-5807-9127
Adam Z Stieg  http://orcid.org/0000-0001-7312-9364
Matteo Pellegrini  http://orcid.org/0000-0001-9355-9564
Atsushi Nakano  http://orcid.org/0000-0001-5702-5039

## Ethics

Human subjects: Human H9 (WiCell) and UCLA4 (UCLA Stem Cell Core) ES cells were grown and differentiated in monolayer in a chemically-defined condition (Arshi et al., 2013; Minami et al., 2012; Young et al., 2016). Usage of all the human ES cell lines is approved by the UCLA Embryonic Stem Cell Research Oversight (ESCRO) Committee and the Institutional Review Boards (IRB) (approval #2009-006-04).

Animal experimentation: Wild type and Akita mice were maintained on C57BL/6 background according to the Guide for the Care and Use of Laboratory Animals published by the US National Institute of Health (NIH Publication No. 85-23, revised 1996). Housing and experiments were performed according to the Institutional Approval for Appropriate Care and Use of Laboratory Animals by the UCLA Institutional Animal Care and Use Committee.(Protocol #2008-127-07).

## Decision letter and Author response

Decision letter https://doi.org/10.7554/eLife.29330.029
Author response https://doi.org/10.7554/eLife.29330.030

# Additional files

## Supplementary files

• Transparent reporting form
DOI: https://doi.org/10.7554/eLife.29330.021

## Major datasets

The following datasets were generated:

| Author(s) | Year | Dataset title | Dataset URL | Database, license, and accessibility information |
|---|---|---|---|---|
| Atsushi N, Matteo P | 2017 | The global transcriptome analysis in the time course of hESC-derived cardiac differentiation | http://www.ncbi.nlm.nih.gov/geo/query/acc.cgi?acc=GSE84815 | Publicly available at the NCBI Gene Expression Omnibus (accession no: GSE84815) |
| Atsushi N, Matteo P | 2017 | Glucose inhibits cardiac maturation through nucleotide | http://www.ncbi.nlm.nih.gov/geo/query/acc.cgi?acc=GSE84814 | Publicly available at the NCBI Gene Expression Omnibus (accession no: GSE84814) |

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
