## [Decision Letter]

Thank you for submitting your article "Glucose inhibits cardiac muscle maturation through nucleotide biosynthesis" for consideration by *eLife*. Your article has been reviewed by three peer reviewers, and the evaluation has been overseen by a Reviewing Editor and Sean Morrison as the Senior Editor. The following individuals involved in review of your submission have agreed to reveal their identity: Chuck Murry (Reviewer #2).

The reviewers have discussed the reviews with one another and the Reviewing Editor has drafted this decision to help you prepare a revised submission..

Summary:

In this manuscript, Nakano et al. study the role that glucose metabolism plays in cardiomyocyte maturation. They report that glucose starvation enhances multiple indices of maturity in hESC-derived cardiomyocytes, and they identify the PPP and nucleotide biosynthesis pathways as key components of the glucose effect. They demonstrate that glucose uptake in the heart diminishes markedly prenatally, prior to the conversion to fatty acid metabolism, and they provide evidence that maternal hyperglycemia delays cardiomyocyte maturation in utero. They therefore conclude that the control of nucleotide bioavailability is a key regulator of cardiomyocyte maturation that may play a role in congenital heart disease associated with maternal diabetes.

The maturation of stem cell derivatives such as cardiomyocytes is under intense study, and metabolism has emerged as an important regulator. The identification of nucleotide biosynthesis as a key regulator of maturation was surprising and significantly advances our understanding. Overall, this manuscript is significant, and the experiments appear to have been performed carefully. A few key additions and adjustments (as described below) will strengthen the message of the manuscript and enhance its impact.

Essential revisions:

1) The authors used three different types of parameters to evaluate cardiomyocyte maturation: mRNA expression of cardiac markers, mitochondrial function, and calcium transients. Thus, maturation is well described at the molecular and metabolic levels, but less well at the structural and physiological level. While it is possible that this is an intervention that advances maturation on all fronts, it seems equally possible that different subnetworks might be regulated differently. The authors' conclusions regarding cardiomyocyte maturation will be better supported with a more comprehensive evaluation of additional indices of maturity. Structurally, it would be helpful to see assessment of cell size and myofibril content/maturity. Physiologically, it would be helpful to evaluate spontaneous beating rate, measurements of contractile strength (e.g. via traction force microscopy, although other ways would work also), and action potential. Finally, since the authors' RNAseq studies should have yielded data on ssTnI to cTnI isoform switching, down-regulation of HCN4, and up-regulation of KCNJ2, it would be of interest to present these data as additional indices of maturation.

2) The authors argue that nucleotide deprivation is a key mechanism in driving cardiac maturation. This conclusion would be more convincing if, rather than using chemical inhibitors of nucleotide biosynthesis, the authors used siRNA knock down of rate limiting enzymes (or even CRISPR/Cas9 to delete the relevant enzymes, unless such deletions are lethal) to further solidify their model.

3) The maximum OCRs in the high glucose cells are unusually low. Other groups, using 25mM glucose conditions, have seen much greater induction of O_2_ consumption after uncoupling mitochondria. In the experiments presented here, OCR does not increase above the basal rate. This reduces confidence in the veracity of the flux measurements. Can the authors address this discrepancy?

4) It is possible that the glucose-starved cells are energy-starved or otherwise stressed. Did the authors measure ATP levels? Is there cell death? Are various stress pathways or autophagic pathways activated?

5) Increased PGC1a expression level and the MitoTracker staining assay suggest that the 0mM glucose condition stimulated mitochondrial biogenesis. Could the authors perform a mtDNA to nDNA ratio assay to see whether 0mM glucose stimulates mitochondrial biogenesis?

6) Figure 3: It seems that the beating rate was 1 per 10 sec. Please clarify if this was spontaneous or paced. Ideally, this should be paced at a relevant rate.

7) Figure 3—figure supplement 1: Examples of membrane voltage traces should be shown.

8) Figure 7: A hallmark of infants of diabetic mothers is cardiac hypertrophy. As such, heart weights and cardiomyocyte size should be examined.

9) Throughout the manuscript, the authors need to add specific information about their statistical testing.

---

## [Author Response]

Essential revisions:1) The authors used three different types of parameters to evaluate cardiomyocyte maturation: mRNA expression of cardiac markers, mitochondrial function, and calcium transients. Thus, maturation is well described at the molecular and metabolic levels, but less well at the structural and physiological level. While it is possible that this is an intervention that advances maturation on all fronts, it seems equally possible that different subnetworks might be regulated differently. The authors' conclusions regarding cardiomyocyte maturation will be better supported with a more comprehensive evaluation of additional indices of maturity. Structurally, it would be helpful to see assessment of cell size and myofibril content/maturity. Physiologically, it would be helpful to evaluate spontaneous beating rate, measurements of contractile strength (e.g. via traction force microscopy, although other ways would work also), and action potential. Finally, since the authors' RNAseq studies should have yielded data on ssTnI to cTnI isoform switching, down-regulation of HCN4, and up-regulation of KCNJ2, it would be of interest to present these data as additional indices of maturation.

As suggested, we examined whether glucose restriction induces hESC-CM maturity at structural, physiological, and genetic levels as follows. All the parameters confirmed the enhancement of the maturity by glucose restriction.

Structural maturity: Cell size was measured by the forward scatter (FSC) from flow cytometry, which is more accurate and unbiased than 2D cell area analysis. Total of 30,000 cell measurements in each group clearly showed a significant increase in the average cell size. Shown in Author response image 1 is a representative histogram and the average of the mode FSC from biological triplicates. This result is now presented in Figure 2. As another parameter, we measured sarcomere length by a-actinin staining (Author response image 1). Unfortunately, sarcomere length did not show a significant difference between glucose 25 vs 0mM culture. This result is presented in Figure 2—figure supplement 1.

Physiological maturity: Contractility was assessed by Digital Image Correlation (DIC) method. Cell video clips were loaded onto MotionGUI program (Huebsch et al., 2015) to acquire contraction/relaxation speed. As presented in Figure 3, both contraction speed and relaxation speed were significantly faster in hESC-CMs cultured without glucose, suggesting an increase in functional maturity of hESC-CMs in glucose-restricted condition. Although not statistically significant, beating rate tends to be decreased in hESC-CMs under glucose-restricted condition. This result is consistent with the developmental observation that the spontaneous beating rate of cardiomyocytes decreases as the immature pacemaker-like (primitive) cardiomyocytes maturate into working myocardium-like cardiomyocyte with less automaticity (Christoffel et al., 2004, Review; many other reviews). Together, these data suggest that glucose restriction enhances cardiomyocyte contractility, thereby promoting the differentiation of immature pacemaker-like cardiomyocytes maturate into working myocyte-like cells.

Genetical maturity: We extracted the data of several cardiac markers from our RNA-seq data. The TNNI2 level was decreased while TNNT2 and IRX4 levels were increased in glucose-restricted condition, indicating the maturation of hESC-CMs. However, levels of TNNI3, HCN4, and KCNJ2 did not change. These data suggest that, even though glucose-restriction induces the hESC-CM maturity, it does not achieve the maturity level of postnatal cardiomyocytes. This result is presented in Figure 2—figure supplement 1.

2) The authors argue that nucleotide deprivation is a key mechanism in driving cardiac maturation. This conclusion would be more convincing if, rather than using chemical inhibitors of nucleotide biosynthesis, the authors used siRNA knock down of rate limiting enzymes (or even CRISPR/Cas9 to delete the relevant enzymes, unless such deletions are lethal) to further solidify their model.

We took advantage of our 2D culture system to achieve 50-80% knockdown by RNAi targeting 4 key enzymes in glucose metabolism (HK1, RRM2, RRM2B, G6PD and PFKM). The results indicate that inhibition of nucleotide biosynthesis significantly increased the level of TNNT2 (Figure 4—figure supplement 2), supporting the results of chemical inhibitor experiments in Figure 5 and Figure 5—figure supplement 1. The reason why the level of TNNT2 is not as impressive as that of chemical inhibitor assays is possibly because the knockdown efficiency never exceeds 90%: the cells that escaped from knockdown, even though a minority at the beginning, can selectively grow and blunt the outcome after 7-10 days.

3) The maximum OCRs in the high glucose cells are unusually low. Other groups, using 25mM glucose conditions, have seen much greater induction of O2 consumption after uncoupling mitochondria. In the experiments presented here, OCR does not increase above the basal rate. This reduces confidence in the veracity of the flux measurements. Can the authors address this discrepancy?

We appreciate the careful review of the manuscript. The OCR presented in the original manuscript was too small, which we did not notice at the initial submission. Therefore, we repeated the same experiment and obtained a reasonable level of OCR. To understand why the original data showed small OCR, we thoroughly checked the raw data and realized a miscalculation of the protein concentration by 10-fold in the original data. Thus, the original data and the repeated experiments are both consistent with the published data. We are sorry for the confusion. The data shown in the revised Figure 3 include all the OCR measurements including new results and the initial results with corrected calculation. Note that the base-line mitochondrial respiration is not significantly different any longer (Figure 3). However, this does not affect our main conclusion that glucose restriction enhances the maturity of hESC-CMs, because ATP-linked resp (Figure 3) and max mito resp capacity (Figure 3) are still significantly higher in glucose-restricted condition.

4) It is possible that the glucose-starved cells are energy-starved or otherwise stressed. Did the authors measure ATP levels? Is there cell death? Are various stress pathways or autophagic pathways activated?

No, they are not energy-starved. ATP level is not significantly decreased in glucose-restricted condition (Figure 4). In addition, gene ontology analyses from our RNA-seq data identified no significant enrichment of genes associated with stress pathways. No other data including immunostaining, flow cytometry, and video clips suggest an increase in cell death when glucose depletion was started at day 14 or later. Glucose depletion did slow down the cell proliferation (Figure 4), but the cells still grow and differentiate. Thus, glucose is not critical for ATP production, and glucose-derived ATP does not seem to be the limiting factor for the cell growth or differentiation in hESC-CMs. As discussed in the manuscript, glucose is perhaps not about the energy but about the supply of building blocks to the cardiomyocytes.

5) Increased PGC1a expression level and the MitoTracker staining assay suggest that the 0mM glucose condition stimulated mitochondrial biogenesis. Could the authors perform a mtDNA to nDNA ratio assay to see whether 0mM glucose stimulates mitochondrial biogenesis?

In response to this comment, we performed mtDNA/nDNA ratio assay, and found that mtDNA content is significantly higher in glucose-restricted condition (Figure 2), supporting the data of PGC1 expression and MitoTracker staining.

6) Figure 3: It seems that the beating rate was 1 per 10 sec. Please clarify if this was spontaneous or paced. Ideally, this should be paced at a relevant rate.

Thank you for raising this. The original Ca^2+^ transient assay was performed in non-temperature-controlled condition, and the beating rate became slower by the time they were transferred to the confocal microscope. In the revised manuscript, we present the Ca^2+^ transient assay data under temperature-controlled environment, in which the spontaneous beating rate was ~0.5Hz in both G25 and G0 conditions. Under this condition, the V_max_ (ΔF/F0/sec) was still significantly higher in glucose-restricted hESC-CMs, supporting our main claim that glucose inhibits cardiac maturation.

7) Figure 3—figure supplement 1: Examples of membrane voltage traces should be shown.

Please find the representative voltage traces in Figure 3.

8) Figure 7: A hallmark of infants of diabetic mothers is cardiac hypertrophy. As such, heart weights and cardiomyocyte size should be examined.

A hallmark of the congenital heart disease associated with diabetic pregnancy is asymmetric cardiac hypertrophy. As suggested, we conducted the phenotypic analyses of the P1 neonatal hearts from diabetic Akita mouse. Our new data suggest that the average heart weight/body weight was not significantly higher in pups from diabetic mothers (Author response image 2). However, the heart from diabetic mothers often showed asymmetric hypertrophy (Figure 7). The LV and RV free wall thicknesses (defined as the maximal wall thickness at the level of maximal heart diameter) were significantly higher in pups from diabetic mothers. In addition, the size of the neonatal cardiomyocytes from diabetic mothers was significantly smaller by flow cytometry analyses (Figure 7). This observation is consistent with our in vitro data (Figure 2), suggesting that hyperproliferation underlies the congenital heart disease associated with diabetic pregnancy. The original Figure 7 are replaced by these new data.

**Author response image 2. respfig2:** 

9) Throughout the manuscript, the authors need to add specific information about their statistical testing.

Please find the specifics of the statistical testing in the Materials and methods section and Figure Legends.